

# Effects of supplemental lighting with different light qualities on growth and secondary metabolite content of *Anoectochilus roxburghii*

Wei Wang[1,2,3], Minghua Su[1,3], Huihua Li[3], Biyu Zeng[3], Qiang Chang[3] and Zhongxiong Lai[1,2]

[1] College of Horticulture, Fujian Agricultural and Forestry University, Fuzhou, Fujian, China
[2] Institute of Horticultural Biotechnology, Fujian Agricultural and Forestry University, Fuzhou, Fujian, China
[3] Fujian Key Laboratory of Physiology and Biochemistry for Subtropical Plant, Fujian Institute of Subtropical Botany, Xiamen, Fujian, China

Corresponding author
Zhongxiong Lai, Laizx01@163.com

## ABSTRACT

**Background:** *Anoectochilus roxburghii* is a widespread herbaceous plant with high medicinal value. Wild *A. roxburghii* resources face extinction due to their slow growth rate and over exploitation. The growing market demand has led to advances in the field of artificial planting of *A. roxburghii*. Methods to increase the economic benefits of cultivation and the production of medicinal ingredients are very useful.

**Methods:** *A. roxburghii* was exposed to red light, blue light (BL), yellow light (YL), green light, or white light as supplemental lighting at night (18:00–02:00) in a greenhouse or were left in darkness (control, CK) to investigate the effects of various light qualities on growth indices, photosynthetic pigments, chlorophyll fluorescence, root vitality, stomatal density, soluble proteins, sugars, and the accumulation of secondary metabolites.

**Results:** Supplementation of BL had a positive effect on *A. roxburghii* growth and secondary metabolite accumulation. Leaf number, stem diameter, fresh weight, dry weight, chlorophyll a content, and secondary metabolite (total flavonoids, total polyphenols) content increased significantly. YL treatment showed significantly higher soluble sugar and polysaccharide contents than the control.

**Discussion:** BL treatment was conducive to promoting the growth and accumulation of secondary metabolites (total flavonoids, total polyphenols); YL treatment significantly increased the content of soluble sugar and polysaccharides more than the control. Polysaccharides and total flavonoids are important medicinal ingredients of *Anoectochilus*, so future research will focus on the combination of blue and YL.

## INTRODUCTION

*Anoectochilus roxburghii* (Orchidaceae) is a valued perennial herb that is used for medicinal and ornamental purposes in China and many Asian countries. The chemical

composition of *A. roxburghii* includes flavonoids, polysaccharides, alkaloids, amino acids, trace elements, organic acids, cardiac glycosides, and steroids (*Han, Yang & Jin, 2008*). Pharmacological studies have indicated that *A. roxburghii* has unique medicinal properties, including antihyperglycemic (*Shih, Wu & Lin, 2002*; *Cui et al., 2013*; *Zhang et al., 2015*), hepatoprotective (*Zeng et al., 2016*, *2017*; *Yang et al., 2017*), antitumor and immunostimulating effects (*Tseng et al., 2006*), renal protection (*Li et al., 2016*), vascular protection (*Liu et al., 2013*), and antioxidant activity (*Shao et al., 2014a*). Various nutraceutical health products have been produced from *A. roxburghii* because of its healthcare effects. Nevertheless, wild resources of *A. roxburghii* face extinction due to overexploitation, low seed germination rate, slow growth rate, specific growth conditions, and increasingly severe environmental problems (*Shao et al., 2014b*). Artificial cultivation is an effective way to meet the growing market demand for *A. roxburghii* and to avoid the extinction of the wild resources.

Many studies indicate that light has a positive effect on the growth and accumulation of plant secondary metabolites (*Agati et al., 2011*; *Koyama et al., 2012*; *Liu et al., 2015*). Light intensity is one of the important factors of light conditions. *A. roxburghii* belongs to the sciophyte family. Shading is necessary for the normal growth of *A. roxburghii*, which has adapted to shade conditions through increased levels of chloroplasts, grana, and grana lamellae, and higher peroxidase and superoxidase activities (*Shao et al., 2014b*). *Anoectochilus* is suitable for cultivation under low light intensity a photosynthetic photon flux of 30–50 $\mu$mol·m$^{-2}$·s$^{-1}$ for both growth and the production of secondary metabolites (total flavonoids) (*Ma et al., 2010*). In addition to light intensity, light quality is one of the important factors of light conditions too; the visible light spectrum (400–700 nm), especially blue light (BL; 400–500 nm) and red light (RL; 600–700 nm) have the greatest impact on plant growth because they are the major energy sources for photosynthetic $CO_2$ assimilation in plants, and as a signal received by photoreceptors it regulates growth, differentiation, and plant secondary metabolism (*Wang et al., 2001*). In instance, the height of *Salvia miltiorrhiza* Bunge and *Tagetes erecta* L. was higher under BL treatment than under RL or fluorescent white light (WL) treatments (*Heo et al., 2002*). Whereas, *Manivannan et al. (2015)* found that RL had the strongest stimulatory effect on the weight and height of *Rehmannia glutinosa* (Gaertn.) DC. Maximal total flavonoid content and total phenolic content was observed when *Prunella vulgaris* was grown under BL (*Fazal et al., 2016*). However, the highest flavonoids and chlorogenic acid content of Tartary buckwheat sprouts were observed with RL + BL and WL (*Seo et al., 2015*).

*Anoectochilus* is sensitive to light, many studies about light intensity of *Anoectochilus* have been reported, but the effects of light-emitting diode (LED) light quality on the yield and bioactive substance content of *A. roxburghii* have not been studied. Thus, the hypothesis of this study was that supplemental lighting with different LED light qualities at night has a positive effect on the growth and secondary metabolism of *A. roxburghii*. This study aimed to use energy-saving LED lamps to explore the effects of supplemental lighting with different light qualities on growth, photosynthetic pigments, chlorophyll fluorescence, root vitality, stomatal density, soluble protein, sugars, and accumulation of secondary metabolites in the cultivation stage of *A. roxburghii* and to determine

the optimal light quality to increase the medicinal value and economic benefits for *Anoectochilus* growers.

## MATERIALS AND METHODS

### Plant materials and growth conditions

Test species: *A. roxburghii* cv. Narrow-leaf.

The experiment was performed in December of 2016 in the greenhouse of the Fujian Institute of Subtropical Botany, China (24°26′N, 118°04′E). The day (highest)/night (lowest) temperatures of the greenhouse were 23/12 °C. *A. roxburghii* plantlets were obtained from Diyuan Biotechnology Co., Ltd. (Xiamen, China). Tissue-culture bottles were opened to acclimate the plantlets to the environmental conditions of the greenhouse for 15 days. The average fresh weight of the acclimated plantlets was 0.55 g and leaf number was 3.8. These plantlets were transplanted to plastic trays (50 × 30 × 5 cm) at a density of 110 plants per tray. The cultivation soil was composed of peat moss and perlite in a 2:1 ratio by volume.

In the supplemental lighting experiment, natural light was used during the day and LED lamps were used as supplemental lighting at night. LED lamps were provided by Xiamen Guangpu Electronics Co., Ltd. (Xiamen, China). Plantlets were subjected to five different light quality treatments: RL (580–660 nm, peak wavelength 630 nm), BL (440–540 nm, peak wavelength 469 nm), yellow light (YL; 540–620 nm, peak wavelength 592 nm), green light (GL; 460–600 nm, peak wavelength 519 nm), WL (420–780 nm) and a control without supplemental lighting (CK). The spectral characteristics of the lamps are shown in Fig. 1, as measured by a MK350S spectrometer (United Power Research Technology Corporation, Zhunan, Taiwan). Each treatment consisted of one tray with three replications (3 × 110 plants for each treatment). The photoperiod of supplemental lighting was 8 h day$^{-1}$ (18:00–02:00). *A. roxburghii* grown under different light qualities at night are shown in Fig. 2.

The LED lamps were placed above of the plantlets. The conclusion from *Ma et al. (2010)* was referenced and the height of each light fixture was adjusted to ensure the light intensity was 30 ± 1 μmol·m$^{-2}$·s$^{-1}$ (light intensity was measured with a LI-250A light meter (LI-COR Biosciences, Lincoln, NE, USA)). Different treatments were insulated from one another by black shading materials.

### Determination of the physiological and biochemical indexes

After 40 days of treatment, 40 plantlets were chosen randomly from each repeat for subsequent testing. A total of nine of these 40 plantlets were selected randomly for biomass analysis within each treatment. And six of these 40 plantlets were selected randomly for root activity analysis and stomata observation within each treatment. The others were stored in a −80 °C refrigerator after flash freezing.

### Growth and biomass parameter analysis

The leaf numbers, length and width of each leaf, stem diameter, stem length, root length, root numbers, fresh weight, and dry weight were separately determined. The fresh weight of
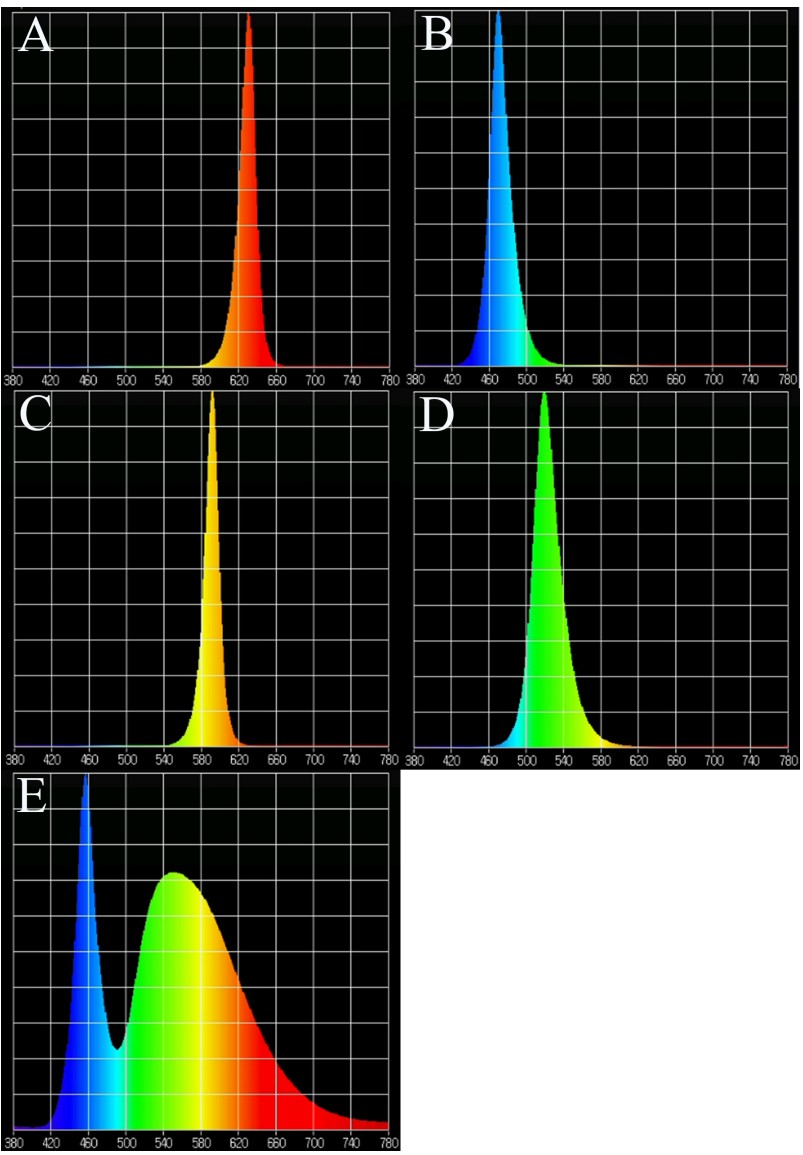

**Figure 1 Light spectra of supplemental lighting with various light qualities: (A) red light; (B) blue light; (C) yellow light; (D) green light; (E) white light.**

the plantlets were measured with an electronic balance (Sartorius, Hamburg, Germany), and the plantlets were dried to a constant weight at 80 °C to determine their dry weight. The stem diameter was measured with a vernier caliper (Tricle Brand Tools, Shanghai, China). The stem length was measured from the top of the plantlet to the base of the first root. The root length was measured from the top of the root tip to the main stem base.

## Chlorophyll content

Chlorophyll content was determined following the method from *Yuan et al. (2010)*. Fresh mature leaves (Half-gram) were collected from each treatment for the determination of chlorophyll content (Chl a, Chl b, Chl a + b,). Samples were ground in a mortar. Chlorophylls were extracted in 15 ml acetone–water solvent (80%, v/v)
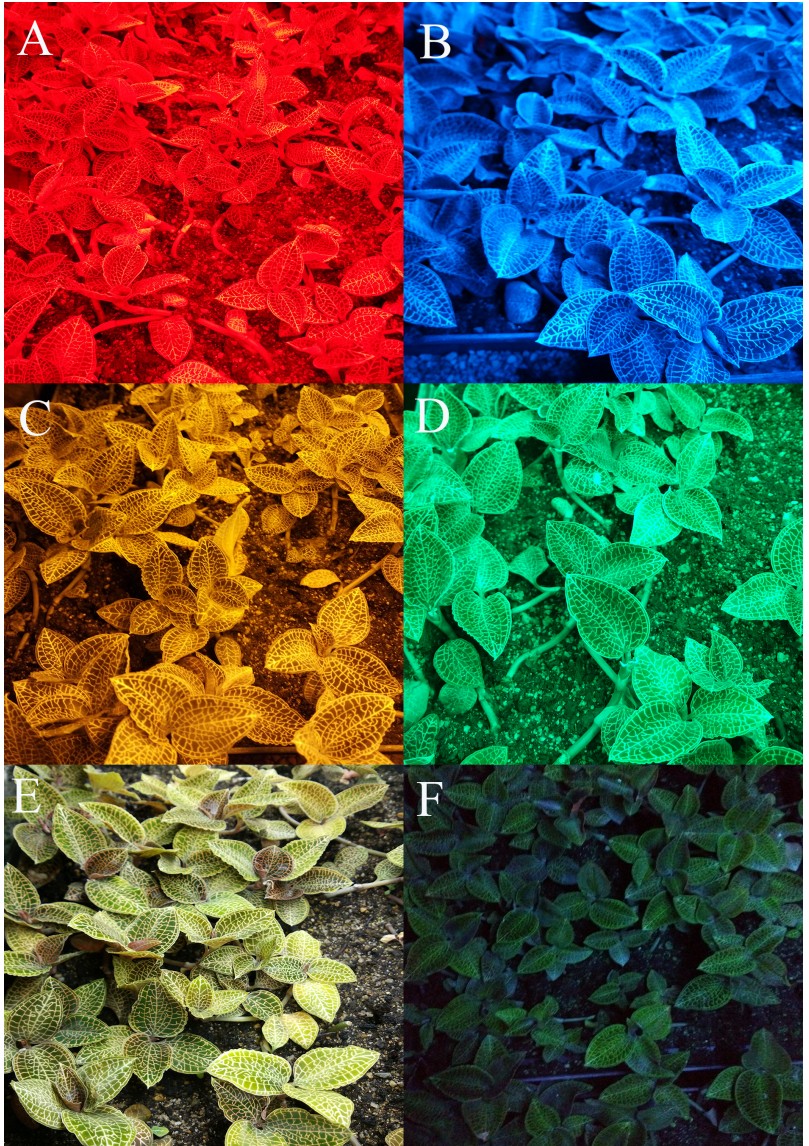

**Figure 2 Cultivation of *A. roxburghii* grown under supplemental lighting with various light qualities: (A) red light; (B) blue light; (C) yellow light; (D) green light; (E) white light; (F) control for 40 days.** The age of *A. roxburghii* plants (175 days) = 120 days (in vitro) + 15 days (acclimation) + 40 days (supplemental lighting process). Photo credit: Wei Wang.

(Xilong Scientific Co., Ltd., Shantou, China) in the dark at room temperature. Chlorophyll quantification was measured at 665 and 649 nm. The results are expressed as milligrams of chlorophyll mass per gram of fresh weight.

## Chlorophyll fluorescence

Chlorophyll fluorescence was measured with a MINIPAM fluorometer (Walz, Effeltrich, Germany) (*Schreiber et al., 1997*). Leaves with consistent maturity were light adapted for approximately 15 min prior to measurements of the effective quantum yield of photochemical energy conversion (Yield) and photochemical (qP) and

nonphotochemical (qN) quenching of chlorophyll fluorescence. The effective quantum yield of qP energy conversion at steady-state photosynthesis was calculated as Yield = (Fm′ − Fs)/Fm′, qP was calculated as (Fm − Fm′)/(Fm′−F$_0$), and qN was calculated as (Fm′−F$_0$′)/(Fm−F$_0$).

## Soluble sugar, reducing sugar, and polysaccharide

Fresh leaves (0.2 g) were homogenized in liquid nitrogen and soluble sugar was extracted in boiling ultrapure water (15 ml) for 20 min. Following centrifugation at 15,000×$g$ for 5 min, the residues were re-extracted under the same conditions. The respective extracts and re-extracts were pooled before analysis. All samples were extracted in duplicate.

The soluble sugar content was evaluated using the anthrone colorimetric method (*Li, 2000*, 194–197). The extracts (0.5 ml) were incubated with the anthrone reagent (five ml) (Shanghai Macklin Biochemical Co., Ltd., Shanghai, China) in a boiling water bath (10 min) to yield a blue-green color and after cooling, its optical density was measured at 620 nm. Standard solutions of glucose (Xilong Scientific Co., Ltd., Shantou, China) were prepared in ultrapure water at a concentration of 10 μg ml$^{-1}$.

The reducing sugar content was evaluated by 3,5-dinitrosalicylic acid (DNS) colorimetry (*Li, 2000*, 197–199). The extracts (0.5 ml) were incubated with the DNS color solution (five ml) (Sinopharm Chemical Reagent Co., Ltd., Shanghai, China) in a boiling water bath (5 min) and after cooling, the optical density was measured at 540 nm. Standard solutions of glucose (Sinopharm Chemical Reagent Co., Ltd., Shanghai, China) were prepared in ultrapure water at a concentration of one mg ml$^{-1}$.

Polysaccharide content = Soluble sugar content − Reducing sugar content

## Stomata observations

Pretreatment of samples was performed based on the method of *Li, Tang & Xu (2013)*. Fully expanded leaves were chosen from each plantlet to observe the stomata. The abaxial and adaxial surfaces of the leaves were wiped with wet absorbent cotton fiber. Then, transparent nail polish was smeared on the two sides of the leaves. After the nail polish had air-dried and formed a membrane, transparent adhesive tape was pressed onto each leaf and was subsequently stripped off. The transparent adhesive tape was then pressed on a slide, which was treated with a neutral plastic seal and made into a temporary slide. The slides were imaged using a Leica DMI 3000B microscope (Leica Microsystems, Wetzlar, Germany). The length, width, and density of stomata were measured with Leica LAS Image Analysis software (Leica Microsystems, Wetzlar, Germany). Stomatal area = Length × Width × 3.14 × 1/4 (μm$^2$). Stomatal density = Number/Field/Field area.

## Root activity

Root activity was determined using the triphenyltetrazolium chloride method (*Li, 2000*, 119–120). A total of 0.3 g samples of fresh roots were treated with five ml of 0.01M 2,3,5-triphenyltetrazolium chloride (TTC) (Sinopharm Chemical Reagent Co., Ltd.,

Shanghai, China) and five ml of 0.07M potassium phosphate buffer (PBS) (Sinopharm Chemical Reagent Co., Ltd., Shanghai, China) for 2 h at 37 °C. The reaction was terminated with two ml of 1M sulfuric acid (Xilong Scientific Co., Ltd., Shantou, China), and the roots were removed and rinsed two to three times with distilled water. The samples were subsequently placed in a mortar with quartz sand (0.3 g) with 10 ml of acetone (Xilong Scientific Co., Ltd., Shantou, China) and ground until the root turned white. The optical density was then measured at 485 nm. To make a standard curve, 0.25, 0.50, 1.00, 1.50, or 2.00 ml of 0.01M TTC was added to five volumetric flasks, and sodium thiosulfate (Xilong Scientific Co., Ltd., Shantou, China) and distilled water were added to reach a volume of 10 ml.

## Soluble protein

The soluble protein content was measured following the method of *Bradford (1976)*. A total of 0.2 g of fresh leaf samples were ground in a mortar with liquid nitrogen to which five ml of 0.07M PBS (Sinopharm Chemical Reagent Co., Ltd., Shanghai, China) was added, followed by centrifugation at $5,000 \times g$ for 10 min. The supernatant was saved and then one ml of extract and five ml of coomassie brilliant blue G-250 (Sinopharm Chemical Reagent Co., Ltd., Shanghai, China) were thoroughly mixed. To generate a standard curve, 0, 0.2, 0.4, 0.6, 0.8, or one ml of 100 g $l^{-1}$ of bovine serum albumin (Sinopharm Chemical Reagent Co., Ltd., Shanghai, China) was added to six volumetric flasks, and distilled water was added to reach a total volume of one ml. The optical density was measured at 595 nm.

## Total flavonoids

For total flavonoids estimation, the method of *Marinova, Ribarova & Atanassova (2005)* was followed with slight modification. Half-gram samples of fresh leaves were homogenized in liquid nitrogen and total flavonoids were extracted in 10 ml of ethanol–water solvent (60%, v/v) (Xilong Scientific Co., Ltd., Shantou, China) in the dark (3 h) and then centrifuged at 15,000 rpm for 10 min. The supernatant (one ml) was mixed with a 0.73M sodium nitrite solution (0.4 ml) (Xilong Scientific Co., Ltd., Shantou, China), a 0.47M aluminum nitrate solution (0.4 ml) (Xilong Scientific Co., Ltd., Shantou, China), 1M sodium hydroxide (four ml) (Guangdong Guanghua Sci-Tech Scientific Co., Ltd., Guangzhou, China) and ethanol–water solvent (60%, v/v) (0.2 ml). The mixture was shaken for 12 min, then its optical density was measured at 510 nm. The total flavonoid content was calculated as milligrams of rutin (Shanghai Macklin Biochemical Co., Ltd., Shanghai, China) equivalent per gram of fresh weight.

## Total polyphenols

The total polyphenol content was determined with the Folin–Ciocalteu reagent according to the method of *Zuo et al. (2012)*. For polyphenol estimation, half-gram samples of fresh leaves were homogenized in liquid nitrogen and total polyphenols were extracted in 15 ml of distilled water (20 min) and then centrifuged at 15,000 rpm for 15 min. The supernatant (one ml) was mixed with Folin–Ciocalteau reagent (1.5 ml) (Sinopharm

Chemical Reagent Co., Ltd., Shanghai, China) and 0.94M sodium carbonate (four ml) (Xilong Scientific Co., Ltd., Shantou, China). The mixture was incubated in the dark for 2 h at room temperature. Its optical density was then measured at 765 nm, and the total polyphenol content was calculated as milligrams of gallic acid (Shanghai Macklin Biochemical Co., Ltd., Shanghai, China) equivalent per gram of fresh weight.

### Statistical analysis

SPSS Version 13.0 (SPSS Inc., Chicago, IL, USA) was used for all statistical analysis. Data was analyzed by one factor, light quality. The results were analyzed by one-way analysis of variance. Duncan's multiple range test was employed to detect differences between means (with $P$ set to 0.05).

## RESULTS

### Morphological observations

The growth indices of *A. roxburghii* treated with different light qualities on day 40 are summarized in Fig. 3 and Table 1; overall, the BL treatment made *A. roxburghii* plantlets grow robustly. Significantly higher leaf numbers ($5.00 \pm 0.50$ vs. $5.89 \pm 0.60$, $P < 0.05$) were observed under the BL treatment, at 17.8% higher than the control. However, the leaf numbers did not differ significantly between the RL ($5.22 \pm 1.09$), YL ($5.00 \pm 0.87$), GL ($5.11 \pm 0.60$), and WL ($5.33 \pm 0.71$) treatments and the control. The leaf length, leaf width, stem length, and root numbers did not differ significantly between all supplemental lighting treatments and the control. Stem diameter was significantly greater in the BL treatment ($2.38 \pm 0.13$ vs. $2.68 \pm 0.43$ mm, $P < 0.05$) than in the control. Root length of the RL ($5.37 \pm 0.85$ vs. $6.53 \pm 0.86$ cm, $P < 0.01$), BL ($5.37 \pm 0.85$ vs. $6.43 \pm 1.09$ cm, $P < 0.05$), and YL ($5.37 \pm 0.85$ vs. $6.76 \pm 0.56$ cm, $P < 0.01$) treatments was significantly higher than in the control. Fresh weight ($1.35 \pm 0.09$ vs. $1.58 \pm 0.12$ g, $P < 0.01$) and dry weight ($0.15 \pm 0.02$ vs. $0.20 \pm 0.03$ g, $P < 0.01$) of the BL treatment were significantly greater than in the control. However, the RL, YL, GL, and WL treatments had no significant effect on the yield of *A. roxburghii*.

### Photosynthetic pigment contents

As shown in Fig. 4, a significantly higher chlorophyll a content ($0.59 \pm 0.03$ vs. $0.82 \pm 0.07$ mg g$^{-1}$ FW, $P < 0.05$) was observed with the BL treatment, at 38.98% higher than in the Control. In this study, chlorophyll b content was not affected significantly by different light qualities. The chlorophyll a + b content showed a similar trend compared to chlorophyll a. However, different light qualities had no significant effect on the chlorophyll a + b content of *A. roxburghii*. The highest chlorophyll a + b content ($1.19 \pm 0.11$ mg g$^{-1}$ FW) was observed with the BL treatment, at 22.6% higher than in the control ($0.97 \pm 0.03$ mg g$^{-1}$ FW), but this increase was not statistically significant ($P > 0.05$).

### Chlorophyll fluorescence

Different light qualities had variable effects on the chlorophyll fluorescence of *A. roxburghii* (Fig. 5). The YL treatment resulted in a significant increase in yield ($0.47 \pm 0.12$

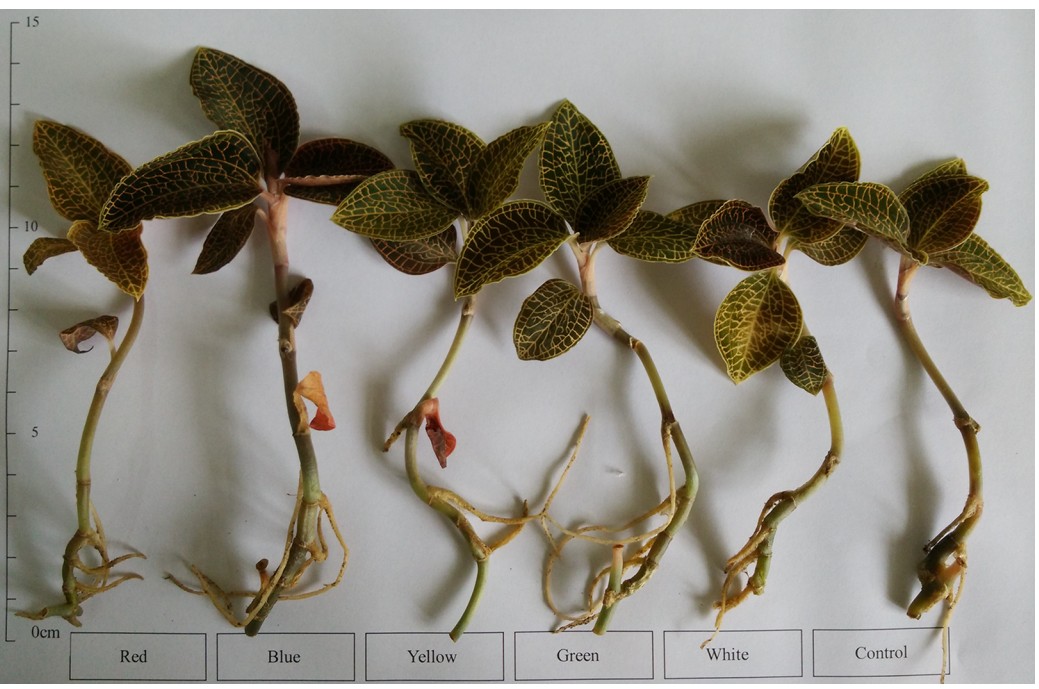

**Figure 3 Appearance of *A. roxburghii* grown under supplemental lighting with different light qualities for 40 days.** The age of *A. roxburghii* plants (175 days) = 120 days (in vitro) + 15 days (acclimation) + 40 days (supplemental lighting process). Photo credit: Wei Wang.

**Table 1 Effects of supplemental lighting with different light qualities on growth indices of *A. roxburghii*.**

| Treatment | Red | Blue | Yellow | Green | White | Control | $F$ ($df_{effect}$, $df_{error}$); $P$ |
|---|---|---|---|---|---|---|---|
| Leaf numbers | 5.22 ± 1.09[a,b] | 5.89 ± 0.60[a] | 5.00 ± 0.87[b] | 5.11 ± 0.60[b] | 5.33 ± 0.71[a,b] | 5.00 ± 0.50[b] | $F_{(5, 48)} = 1.769$; $P = 0.137$ |
| Leaf length (cm) | 3.20 ± 0.40[a] | 3.30 ± 0.43[a] | 3.32 ± 0.66[a] | 3.33 ± 0.27[a] | 3.17 ± 0.19[a] | 3.46 ± 0.30[a] | $F_{(5, 48)} = 0.590$; $P = 0.708$ |
| Leaf width (cm) | 2.19 ± 0.39[a] | 2.34 ± 0.24[a] | 2.26 ± 0.37[a] | 2.26 ± 0.26[a] | 2.12 ± 0.19[a] | 2.39 ± 0.15[a] | $F_{(5, 48)} = 1.117$; $P = 0.364$ |
| Stem diameter (mm) | 2.41 ± 0.20[b] | 2.68 ± 0.43[a] | 2.46 ± 0.35[a,b] | 2.31 ± 0.20[b] | 2.31 ± 0.15[b] | 2.38 ± 0.13[b] | $F_{(5, 48)} = 2.464$; $P = 0.046$ |
| Stem length (cm) | 5.89 ± 0.84[a] | 6.53 ± 1.04[a] | 5.97 ± 0.88[a] | 6.01 ± 0.34[a] | 5.83 ± 0.50[a] | 6.01 ± 0.77[a] | $F_{(5, 48)} = 0.966$; $P = 0.448$ |
| Root numbers | 4.00 ± 0.71[a] | 4.11 ± 0.60[a] | 3.44 ± 0.53[a] | 3.44 ± 0.53[a] | 4.11 ± 0.78[a] | 3.44 ± 0.88[a] | $F_{(5, 48)} = 2.321$; $P = 0.057$ |
| Root length (cm) | 6.53 ± 0.86[a] | 6.43 ± 1.09[a] | 6.76 ± 0.56[a] | 6.06 ± 0.55[a,b] | 5.99 ± 1.17[a,b] | 5.37 ± 0.85[b] | $F_{(5, 48)} = 2.372$; $P = 0.053$ |
| Fresh weight (g) | 1.43 ± 0.22[a,b] | 1.58 ± 0.12[a] | 1.43 ± 0.24[a,b] | 1.29 ± 0.20[b] | 1.41 ± 0.14[a,b] | 1.35 ± 0.09[b] | $F_{(5, 48)} = 2.683$; $P = 0.032$ |
| Dry weight (g) | 0.16 ± 0.03[b] | 0.20 ± 0.03[a] | 0.18 ± 0.03[a,b] | 0.15 ± 0.03[b] | 0.16 ± 0.02[b] | 0.15 ± 0.02[b] | $F_{(5, 48)} = 4.196$; $P = 0.003$ |

Notes:
Values represent mean ± SE of nine replicates; and different letters within a row indicate significant differences at $P < 0.05$. $F$, $P$, and $df$ represent $F$-value, $P$-value, and degree of freedom of the ANOVA, respectively.

vs. 0.62 ± 0.08, $P < 0.05$), at 31.9% higher than in the control. A significant decrease in yield was observed in RL (0.47 ± 0.12 vs. 0.32 ± 0.06, $P < 0.05$) treatment, at 31.9% lower than in the control. However, the yield value did not differ significantly between the BL (0.56 ± 0.05), GL (0.38 ± 0.17), and WL (0.34 ± 0.11) treatments and the control.

There was no significant difference in the qP value between all of the supplemental lighting treatments and the control.

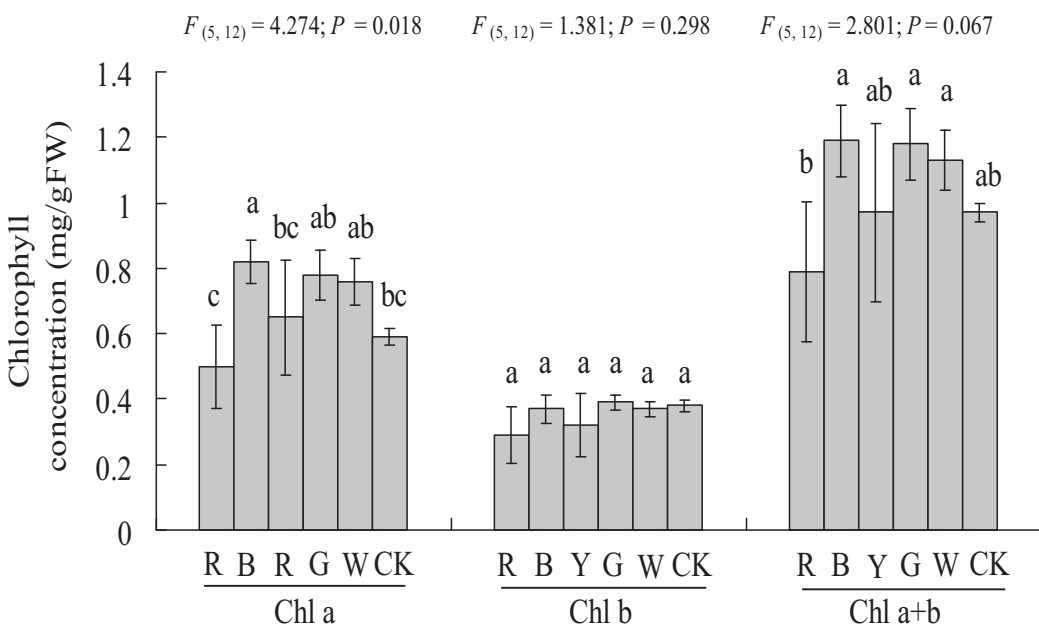

**Figure 4 Effects of supplemental lighting with different light qualities on chlorophyll content of** *A. roxburghii.* Study sites: RL, red light; BL, blue light; YL, yellow light; GL, green light; WL, white light; CK, control. Values represent the mean of three replicates, the bars represent the standard error. In the same row, values marked with different letters are significantly different ($P < 0.05$). *F* and *P* represent *F*-value and *P*-value of the ANOVA, respectively.

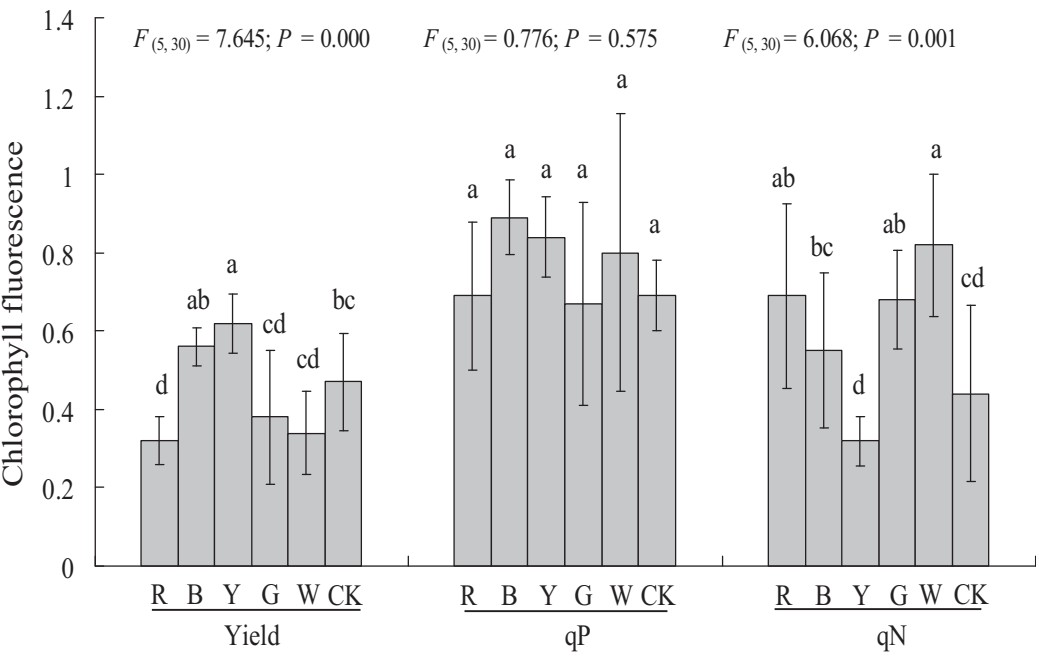

**Figure 5 Effects of supplemental lighting with different light qualities on chlorophyll fluorescence of** *A. roxburghii.* Study sites: RL, red light; BL, blue light; YL, yellow light; GL, green light; WL, white light; CK, control. Values represent the mean of six replicates, the bars represent the standard error. In the same row, values marked with different letters are significantly different ($P < 0.05$). *F* and *P* represent *F*-value and *P*-value of the ANOVA, respectively.

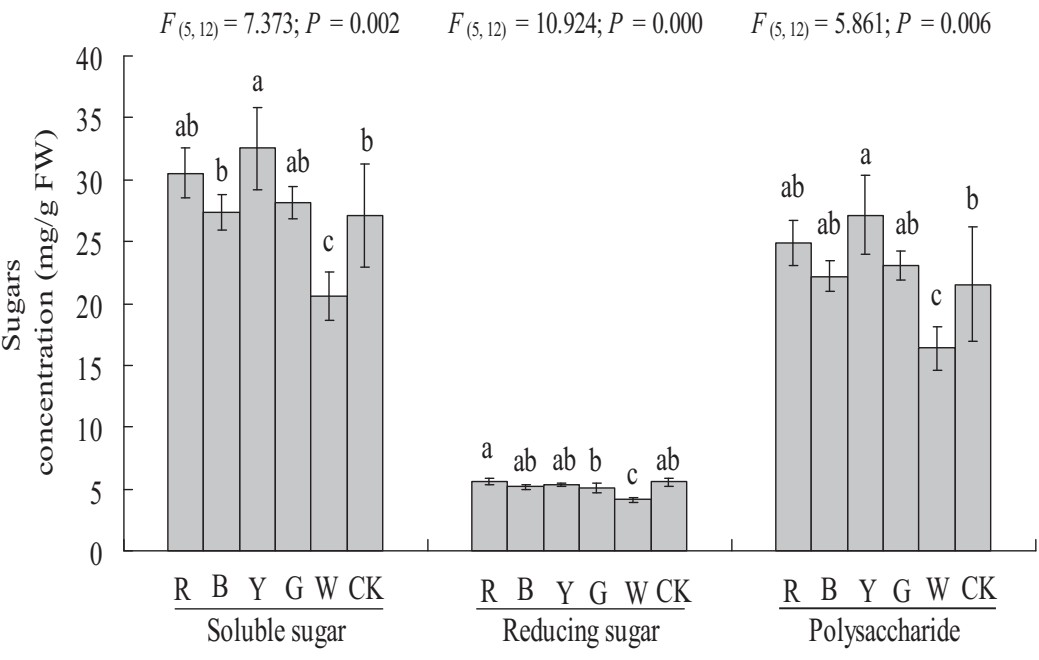

**Figure 6 Effects of supplemental lighting with different light qualities on sugar content of _A. roxburghii._** Study sites: RL, red light; BL, blue light; YL, yellow light; GL, green light; WL, white light; CK, control. Values represent the mean of three replicates, the bars represent the standard error. In the same row, values marked with different letters are significantly different ($P < 0.05$). $F$ and $P$ represent $F$-value and $P$-value of the ANOVA, respectively.

The qN value was significantly higher in the RL treatment ($0.44 \pm 0.22$ vs. $0.69 \pm 0.23$, $P < 0.05$), GL treatment ($0.44 \pm 0.22$ vs. $0.68 \pm 0.13$, $P < 0.05$), and WL treatment ($0.44 \pm 0.22$ vs. $0.82 \pm 0.18$, $P < 0.01$) than in the control. However, the qN value did not differ significantly between the BL ($0.55 \pm 0.20$) and YL ($0.32 \pm 0.06$) treatments and the control.

## Soluble sugar, reducing sugar, and polysaccharide contents

The soluble sugar content was significantly higher in the YL treatment ($27.10 \pm 4.21$ vs. $32.51 \pm 3.32$ mg g$^{-1}$ FW, $P < 0.05$) than in the control. A significantly lower soluble sugar content was observed in the WL treatment ($27.10 \pm 4.21$ vs. $20.53 \pm 1.95$ mg g$^{-1}$ FW, $P < 0.01$) than in the control. The soluble sugar content did not differ significantly between the RL ($30.52 \pm 2.03$ mg g$^{-1}$ FW), BL ($27.38 \pm 1.45$ mg g$^{-1}$ FW), and GL ($28.1 \pm 1.32$ mg g$^{-1}$ FW) and control treatments ($27.10 \pm 4.21$ mg g$^{-1}$ FW) (Fig. 6).

The reducing sugar content did not differ significantly between the RL ($5.62 \pm 0.26$ mg g$^{-1}$ FW), BL ($5.17 \pm 0.22$ mg g$^{-1}$ FW), YL ($5.37 \pm 0.13$ mg g$^{-1}$ FW), GL ($5.04 \pm 0.39$ mg g$^{-1}$ FW) or control ($5.54 \pm 0.38$ mg g$^{-1}$ FW) treatments, but a significantly lower reducing sugar content was observed in the WL treatment ($5.54 \pm 0.38$ vs. $4.14 \pm 0.22$ mg g$^{-1}$ FW, $P < 0.01$) (Fig. 6).

The polysaccharide content was significantly higher in the YL treatment ($21.55 \pm 4.58$ vs. $27.13 \pm 3.19$ mg g$^{-1}$ FW, $P < 0.05$), at 25.9% higher than in the control treatments. A significantly lower polysaccharide content was observed in the WL treatment

$(21.55 \pm 4.58$ vs. $16.38 \pm 1.74$ mg g$^{-1}$ FW, $P < 0.05$). However, the polysaccharide content did not differ significantly between the RL ($24.90 \pm 1.80$ mg g$^{-1}$ FW), BL ($22.21 \pm 1.29$ mg g$^{-1}$ FW), GL ($23.06 \pm 1.18$ mg g$^{-1}$ FW) and control ($21.55 \pm 4.58$ mg g$^{-1}$ FW) treatments (Fig. 6).

## Stomatal observation

The stomatal area of *A. roxburghii* grown under different light qualities is shown in Fig. 7A. The stomatal area did not differ significantly between the RL ($1075.07 \pm 143.71$ $\mu$m$^2$), BL ($1016.76 \pm 140.50$ $\mu$m$^2$), YL ($921 \pm 110.54$ $\mu$m$^2$), GL ($961.44 \pm 138.74$ $\mu$m$^2$), and WL ($1051.58 \pm 69.33$ $\mu$m$^2$) treatments and the control ($1003.55 \pm 99.48$ $\mu$m$^2$).

As shown in Fig. 7B, a significantly higher stomatal density value was observed in the YL treatment ($17.9 \pm 2.64$ vs. $23.3 \pm 3.78$ mm$^{-2}$, $P < 0.01$) than in the control. However, the stomatal density did not differ significantly between the RL ($17.4 \pm 2.82$ mm$^{-2}$), BL ($20.4 \pm 3.07$ mm$^{-2}$), GL ($20 \pm 3.83$ mm$^{-2}$), WL ($17.3 \pm 2.42$ mm$^{-2}$) treatments and the control ($17.9 \pm 2.64$ mm$^{-2}$).

## Root vitality

The root vitality was significantly higher in the YL treatment ($0.41 \pm 0.10$ vs. $0.53 \pm 0.06$ mg g$^{-1}\cdot$h$^{-1}$, $P < 0.05$) than in the control. However, the root vitality did not differ significantly between the RL ($0.33 \pm 0.08$ mg g$^{-1}\cdot$h$^{-1}$), BL ($0.35 \pm 0.01$ mg g$^{-1}\cdot$h$^{-1}$), GL ($0.46 \pm 0.02$ mg g$^{-1}\cdot$h$^{-1}$), and WL ($0.41 \pm 0.02$ mg g$^{-1}\cdot$h$^{-1}$) treatments and the control ($0.41 \pm 0.10$ mg g$^{-1}\cdot$h$^{-1}$) (Fig. 7C).

## Soluble protein content

A significantly lower soluble protein content was observed in the BL treatment ($3.05 \pm 0.30$ vs $2.30 \pm 0.28$ mg g$^{-1}$ FW, $P < 0.01$) than in the control. However, soluble protein content did not differ significantly between the RL ($2.83 \pm 0.32$ mg g$^{-1}$ FW), YL ($2.63 \pm 0.09$ mg g$^{-1}$ FW), GL ($3.13 \pm 0.36$ mg g$^{-1}$ FW), and WL ($2.72 \pm 0.17$ mg g$^{-1}$ FW) treatments and the control ($3.05 \pm 0.30$ mg g$^{-1}$ FW) (Fig. 7D).

## Secondary metabolite contents

A significantly higher total flavonoid content was observed in the BL treatment ($1.57 \pm 0.20$ vs $1.95 \pm 0.03$ mg g$^{-1}$ FW, $P < 0.01$), at 24.2% higher than in the control treatment (Fig. 7E). The total flavonoid content did not differ significantly between the RL ($1.74 \pm 0.11$ mg g$^{-1}$ FW), YL ($1.59 \pm 0.08$ mg g$^{-1}$ FW), GL ($1.72 \pm 0.02$ mg g$^{-1}$ FW), WL ($1.56 \pm 0.14$ mg g$^{-1}$ FW) and control ($1.57 \pm 0.20$ mg g$^{-1}$ FW) treatments.

The total polyphenol content was significantly higher in the BL treatment ($3.70 \pm 0.12$ vs $4.77 \pm 0.23$ mg g$^{-1}$ FW, $P < 0.01$) than in control treatment. A significantly lower total polyphenol content was observed in the GL treatment ($3.70 \pm 0.12$ vs. $3.30 \pm 0.54$ mg g$^{-1}$ FW, $P < 0.05$) than in the control. The total polyphenol content did not differ significantly between the RL ($4.38 \pm 0.43$ mg g$^{-1}$ FW), YL ($3.52 \pm 0.04$ mg g$^{-1}$ FW), WL ($4.12 \pm 0.71$ mg g$^{-1}$ FW) and control ($3.70 \pm 0.12$ mg g$^{-1}$ FW) treatment (Fig. 7F).

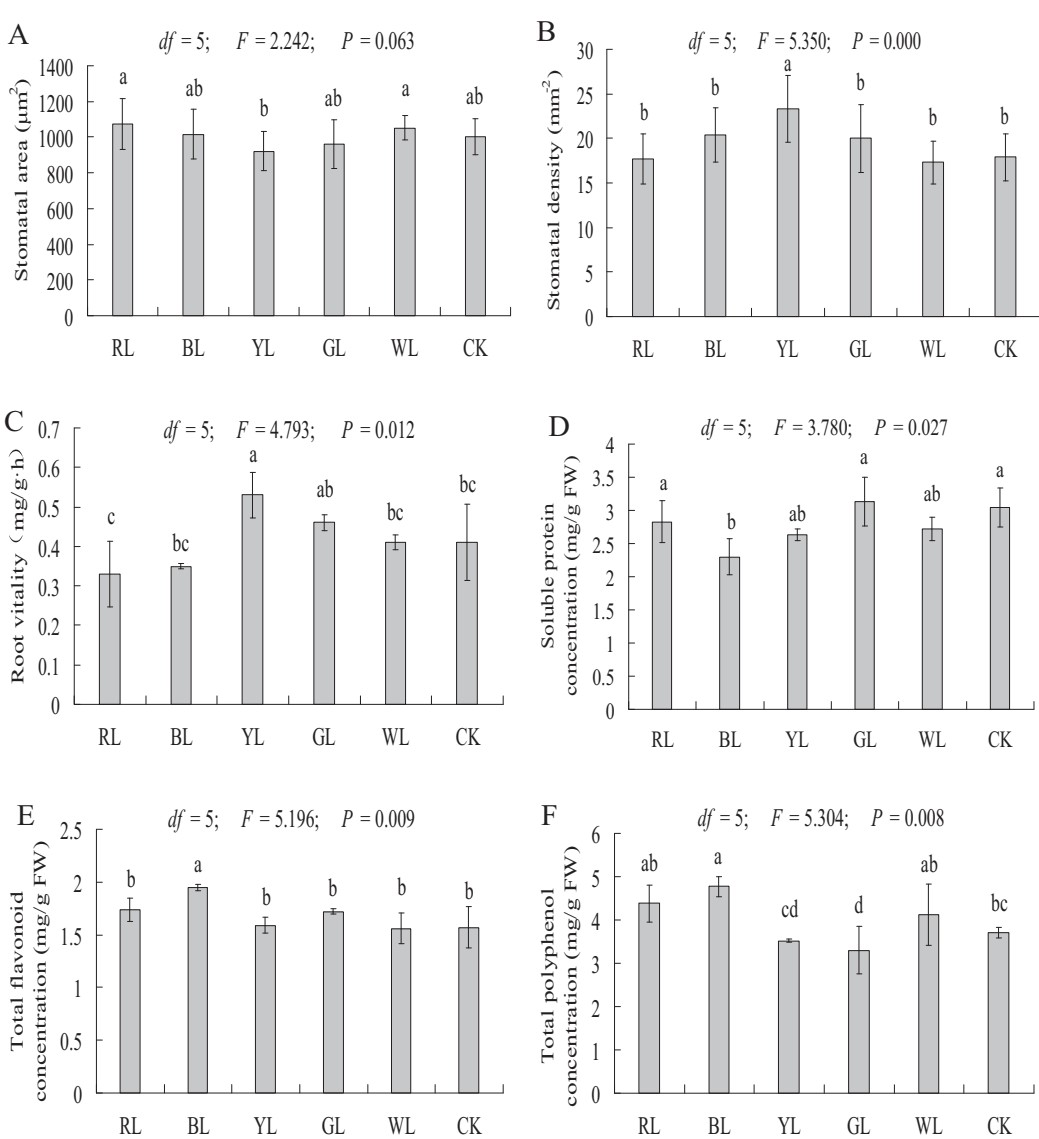

**Figure 7 Effects of supplemental lighting with different light qualities on stomatal area (A), stomatal density (B), root vitality (C), soluble protein (D), total flavonoid content (E), total polyphenol content (F) of *A. roxburghii*.** Study sites: RL, red light; BL, blue light; YL, yellow light; GL, green light; WL, white light; CK, control. Values of stomatal area (A) and stomatal density (B) represent mean of 10 replicates, values of root vitality (C), soluble protein (D), total flavonoid (E), and total polyphenol (F) represent the mean of three replicates, the bars represent the standard error. In the same row, values with the same superscript letter are not significantly different ($P > 0.05$); those with different superscript letters are significantly different ($P < 0.05$). F, P, and df represent F-value, P-value, and degree of freedom of the ANOVA, respectively.

## DISCUSSION

Light quality is an important environmental factor, amongst the light spectra, red and blue wavelengths are the primary spectral wavelengths and highly influence the plant primary and secondary metabolism (*Johkan et al., 2010*). RL is reported to contribute to photosynthetic apparatus development and may increase starch accumulation in several plant species (*Kobayashi, Amore & Lazaro, 2013*). BL is important for

photosynthesis, chloroplast development, chlorophyll formation, and chemical composition of plants (*Hogewoning et al., 2010*).

## Biomass parameter response of *A. roxburghii* to light quality

Biomass is an important indicator in medicinal plants. In the present study, the biomass parameters of *A. roxburghii* differed in their responses to different light qualities, but the BL treatment made *A. roxburghii* plantlets grow robustly. Exposure to the BL treatment significantly increased the leaf numbers, stem diameter, fresh weight, and dry weight of *A. roxburghii*. The root length of the RL, BL, and YL treatments was significantly higher than in the control. There were similar conclusions in other related studies involving, *Stevia rebaudiana* Bertoni (*Simlat et al., 2016*), *Cardamine fauriei* (*Abe et al., 2015*), and *R. glutinosa* (*Manivannan et al., 2015*), where exposure to the BL treatment showed the largest leaf numbers. BL is more effective in enhancing fresh weight and dry matter production in buckwheat sprouts (*Lee et al., 2014*). However, RL had a stimulating effect on leaf numbers and the root length of cucumber seedlings (*Su et al., 2014*). These results indicate that plant species differ in their responses to light quality, but BL generally promotes plant growth and dry matter accumulation.

## Photosynthetic pigment response of *A. roxburghii* to light quality

Chlorophyll content is an important determinant of photosynthetic and dry matter production (*Ghosh et al., 2004*). Studies have confirmed that light, especially BL, plays an important role in the synthesis of plant chlorophyll. Cultured *Phalaenopsis* "Fortune Saltzman" seedlings treated with BL for 5 months showed significantly higher responses in terms of chlorophyll a, chlorophyll b and total chlorophyll (*Anuchai & Hsieh, 2017*). BL contributes positively to chlorophyll synthesis in *S. rebaudiana* Bertoni, and plantlets grown under RL showed the lowest levels of chlorophyll (*Simlat et al., 2016*). *Senger (1982)* documented that BL plays an important role in chloroplast development and the formation of chlorophyll. In our study, BL increased the chlorophyll a content of *A. roxburghii* significantly; Chlorophyll b was not sensitive to light quality. Studies confirm that chlorophyll, especially chlorophyll a, plays an important role in the process of photosynthesis and the accumulation of dry matter (*Naidu et al., 1984*). In our study, the highest dry matter content of *A. roxburghii* was observed in the BL treatment, which illustrates that BL promoted the synthesis of chlorophyll, especially chlorophyll a, thereby promoting photosynthesis and the accumulation of dry matter.

## Chlorophyll fluorescence response of *A. roxburghii* to light quality

Chlorophyll fluorescence is an important signal of photosynthetic synthesis and plant responses to the external environment. In our study, the highest yield and lowest qN were observed with the YL treatment, which indicates that YL can improve the photosynthetic efficiency of *A. roxburghii*. The second-highest dry matter content also confirmed the promotion of YL on photosynthetic synthesis in *A. roxburghii*. There was no significant difference in qP between all treatments and the control. However, different results appear in other similar studies. In *C. acuminate* seedlings, RL increased the efficiency of

photosynthesis, while BL caused photoinhibition (*Yu et al., 2016*). *Wang et al. (2009)* found that YL caused a decrease in the quantum yield of the PSII electron transport compared to *Cucumis sativus* plants grown under WL. These results indicate that plant species differ in their responses to light quality in terms of chlorophyll fluorescence.

### Sugar response of *A. roxburghii* to light quality

As sugars are an important carbon source and osmoregulator of plant growth, sugars levels reflect the photosynthetic capacity of a plant. Light quality is also an important factor affecting the metabolism of sugars in plants. In our study, YL significantly increased the soluble sugar and polysaccharide content of *A. roxburghii*. WL caused a marked decrease in soluble sugars, reducing sugar, and polysaccharides in *A. roxburghii*. The significant increase in carbohydrates under the YL treatment also corresponded to the change in chlorophyll fluorescence, which indicates that YL indeed promoted photosynthesis in *A. roxburghii*. However, the responses to light quality of plants varied among species and cultivars. *Anuchai & Hsieh (2017)* showed that RL promoted sugar accumulation significantly in cultured *Phalaenopsis* seedlings. BL significantly increased the soluble sugar content in *Tobacco* (*Yang et al., 2016*). The total sugar from grape skin was highest in blue LED-treated plants, followed by red LED-treated plants (*Kondo et al., 2014*). The above results indicate that the mechanism of different light qualities regulating the metabolism of carbohydrate in plants is complex and needs further exploration.

### Stomata response of *A. roxburghii* to light quality

Stomata are the pathway for air and water vapor during carbon assimilation, respiration, and transpiration, and their quantity is regulated by the opening and closing of guard cells. Many exogenous factors influence the opening and closing of stomata, such as light, temperature, and $CO_2$. In our study, YL caused a significant increase in stomatal density in *A. roxburghii*, but there was no significant difference in the stomatal area between LED light treated plants and control treated plants. In other similar studies, the stomata of different plants responded differently to light quality. In *S. rebaudiana*, BL not only increased the number of stomata but also caused stomatal opening (*Simlat et al., 2016*). *Macedo et al. (2011)* found that red and blue fluorescent light reduced the number of stomata on the adaxial face and abaxial face of *Alternanthera brasiliana* leaves, respectively. These results indicate that the effects of different light qualities on the stomata of different plants are inconsistent, and the mechanism of light regulating plant stomata needs further research.

### Soluble protein response of *A. roxburghii* to light quality

Soluble protein is an important osmotic regulator and nutrient involved in various metabolic enzymes. Soluble protein content is an embodiment of plant resistance and metabolism. In this study, BL caused a significant decrease in soluble protein levels in *A. roxburghii*, but the soluble protein content did not differ significantly between other treatments and controls. However, the effects of light quality on the soluble protein

content in other plants were different. In *P. vulgaris* L., the soluble protein content was higher under white fluorescent tube lights compared to all colored lights (*Fazal et al., 2016*). The highest total protein content of *A. absinthium* grown under various colored illuminations was observed under RL (*Tariq, Ali & Abbasi, 2014*).

## Secondary metabolite response of *A. roxburghii* to light quality

Secondary metabolism is an indispensable part of plant life and related closely to plant growth and environmental factors. Light is an essential environmental factor that affects the accumulation of secondary metabolites. BL, UV-B, and UV-A were considered to trigger the gene expression of chalcone synthase, the first committed step in the flavonoid metabolic pathway, and effect flavonoid metabolism in Arabidopsis (*Christie & Jenkins, 1996*; *Fuglevand, Jackson & Jenkins, 1996*). In *P. vulgaris* L., BL was effective for total phenolic and flavonoid content (*Fazal et al., 2016*). In Tartary buckwheat sprouts, BL induced the accumulation of flavonoids (*Seo et al., 2015*). The phenolic levels in *Lactuca sativa* L. seedlings treated with BL increased significantly (*Johkan et al., 2010*). *Taulavuori et al. (2016)* suggested that both BL and RL may be needed to regulate the accumulation of phenolics in basil. In our study, significantly higher total flavonoid and polyphenol contents were observed in BL. These results indicate that BL promotes the accumulation of plant secondary metabolites, which provides an effective method to increase the medicinal component contents of medicinal plants.

## CONCLUSION

The supplementation of LED BL had a positive effect on *A. roxburghii*, which achieved greater biomass and a markedly higher content of chlorophyll and secondary metabolites (total flavonoids and total polyphenols) compared to controls. These results suggest that the supplementation of LED BL is a good choice for *A. roxburghii* growers to increase production and secondary metabolite content. This is conducive to meeting the increasing market demand while avoiding the extinction of wild resources of *A. roxburghii*. Considering the promotion of the soluble sugar and polysaccharide contents after the YL treatment, further research will focus on the combination of BL and YL.

## ACKNOWLEDGEMENTS

We thank Xiamen Guangpu Electronics Co., Ltd. for providing LED lamps, and Diyuan (Xiamen) Biotechnology Co., Ltd. for providing *A. roxburghii* plantlets.

### Funding

Financial support for this study was provided by Science and Technology Planning Project of Xiamen City (No. 3502Z20151257), Science and Technology Key Project of Fujian Province (No. 2015NZ0002-1), and Fujian Institute of Subtropical Botany Foundation (No. 20150318-1). The funders had no role in study design, data collection and analysis, decision to publish, or preparation of the manuscript.

## Grant Disclosures

The following grant information was disclosed by the authors:
Science and Technology Planning Project of Xiamen City: No. 3502Z20151257.
Science and Technology Key Project of Fujian Province: No. 2015NZ0002-1.
Fujian Institute of Subtropical Botany Foundation: No. 20150318-1.

## Competing Interests

The authors declare that they have no competing interests.

## Author Contributions

- Wei Wang conceived and designed the experiments, performed the experiments, analyzed the data, contributed reagents/materials/analysis tools, prepared figures, and/or tables, approved the final draft.
- Minghua Su conceived and designed the experiments, authored or reviewed drafts of the paper, approved the final draft.
- Huihua Li contributed reagents/materials/analysis tools, approved the final draft.
- Biyu Zeng contributed reagents/materials/analysis tools, approved the final draft.
- Qiang Chang contributed reagents/materials/analysis tools, approved the final draft.
- Zhongxiong Lai conceived and designed the experiments, authored or reviewed drafts of the paper, approved the final draft.

## Data Availability

The raw data are provided as a Supplemental File.

## Supplemental Information

Supplemental information for this article can be found online at http://dx.doi.org/10.7717/peerj.5274#supplemental-information.

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
