# Peer review of "Effects of supplemental lighting with different light qualities on growth and secondary metabolite content of Anoectochilus roxburghii"

_PeerJ, doi:10.7717/peerj.5274_

## Round 0.1 · original submission · Major Revisions

After carefully evaluating your submission, I was much more critical and negative with the quality of the actual form of your paper than both the reviewers. My decision is that your submission is not suitable for publication in PeerJ in its actual form. However, your research provides valuable results worthy to publish, but it requires an in deep revision and whole rewriting. Thus I encourage you to produce a highly improved version of the manuscript following all the indications by the reviewers and myself, asking to very single suggestion posed by the reviewers in a separate revision notes document. Please use blue font for the next text you add. Your new version will enter into the review process again, and thus this opportunity does not ensure a future acceptation of your submission.

Handling editor notes for the author

The aim of your research is highly specific, focusing directly on your species and your measured variables. In my opinion you should clearly define your research question and put it a broader context, suitable to a general audience in a general scientific journal. Why your question is meaningful, what is your hypothesis, what is the ecological and physiological framework where your hypothesis is sited, identify the particular gap in the knowledge where you set your questions, why that gap is relevant and timely, and how your study is contributing to filling that gap. Otherwise your paper has just the value of a scientific-technical report, which is more suitable for a more specific and technical journal.

The language should be improved for avoiding misunderstanding and please make an effort in improving your writing following the usual scientific style.

The methodology section has a very poor writing for the style that is expected in a scientific journal and need a deep improvement. In instance, please provide a brief explanation of methods used and not just a list of references. Sometimes your manuscript follows a poor draft-style (L77:95). Please provide the manufacturer and brand of your instruments and reagents. Be precise when reporting units (not using % is preferred) and with subjective adjectives such as “moderate sand”. Please clearly state the sample size for each analysis.

For the results section, the full anova table are required, reporting the main effects F, degreed of freedom of N and error and associated P values. Similarly, figures should be edited for improving clarity. And discussion should clearly address what is the physiological basis supporting your results, and how your results contribute to the existing science in the field.

Reviewer 1 ·

Basic reporting

no comment

Experimental design

no comment

Validity of the findings

no comment

Additional comments

The manuscript provides interesting information regarding the use of LEDs as a supplemental lighting in Anoectochilus roxburghii cultivation. The obtained results could be of interest to plant producers and breeders.
Generally, this manuscript is well prepared and meets the scientific standards. I recommend this paper for publication in PeerJ after a major revision. My specific comments/remarks are given below. Some language improvements would also be desirable.

Remarks:
Title page
Please, specify the corresponding author.

Keywords
There is no keywords. Please specify them.

Introduction
The introduction needs to be expanded. What are the findings from previous research with the effect of light on growth and biochemical properties of Anoectochilus roxburghii? What features make it a good choice for this type of research?
In the sentence: “Anoectochilus is a perennial herb that belongs to the genus Anoectochilus (Orchidaceae).”, ‘Anoectochilus’ should be substituted with ‘Anoectochilus roxburghii ‘.

Materials and Methods
Plant materials: Please specify the temperature in the greenhouse. The information ‘12-23°C’ is not precise. Please give the information what temperature was during the day/lighting time and what temperature was during the night.
In line 59 there should be ‘0.55 and 3.8 g’ instead of ‘0.55 g and 3.8’.
In line 60 there should be ‘(50 x 30 x 5 cm)’ instead of ‘(50 cm x 30 cm x 5 cm).
Please specify the photoperiod applied for plantlets growth. Was there applied any other lighting during the day? Please specify (line 69).

Results
Please do not use point after ‘g’ in ‘mg/g.FW’.
There is no reason to write the word ‘yield’ using capital letters ‘YIELD’.

References
This section needs a minor revision; latin names of species should be written in italics; the titles should be written like in a sentence; there is no need to underline names of authors; the full journal titles should be given – please refer to ‘Instructions for Authors’.

Tables
Table 1: Please specify how many replications were used to calculate the means.

Figures
Figures 2 and 3: please include the information about the age of plants.
Figure 4-12: please specify what the given values are, probably means? How many replications were used to calculate the means; what is the meaning of the letters and vertical bars above the columns? Please, specify also the probability.

·

Basic reporting

No comment.

Experimental design

Methods need to be described in more detail, specific comments are provided below.

Validity of the findings

Please see my specific comments regarding how results are discussed below.

Additional comments

lines 65-66: Please describe the light treatments in more detail, as “green” can mean different things according to different definitions. Thus report e.g. GL 500-580 nm and the peak wavelength.
line 67: What is the spectrometer manufacturer, city, country?
line 71: Please change height of each treatment, to height of each light fixture.
line 72: Light intensity between which wavelenghts, PAR?
line 77: Were these nine plants different from the ones above, I presume so, because the other ones you freezed? Or did you first do these measurements and then used the same plants for subsequent testing.
lines 84-95: Methods need to be described in more detail, merely “following the method of” is not enough. Key steps need to be described; how the samples were grinded, were subsamples of leaf part used etc. These need to be full sentences where the reader gets the idea how the measurements were done, similarly how it is now written concerning the total flavonoids measurements.

Results section needs rigorous rewriting. As it is, the results are not reported entirely correctly. Please pay detailed attention to what is different from what. It is very important to always state the treatment to what are you comparing against. Please use the opportunity to report the other way around too, so to say, e.g. that what treatments were similar with the control.
line 127: Similarly to leaf number, please report here the stem diamter for BL treatment and for the treatment it compared with.
line 128: Remove “supplemental lighting treatments caused various degrees of increase”, just report those that were significant here and add to which treatment they were compared with.
lines 129-133: One can not say “except GL”, as the only treatment statistically different from control is BL for both fresh and dry weight, as all the other treatments share the same letter with control too.
lines 136-137: Rephrase, as the content did not change statistically significantly.
line 139: Add to the end of the sentence “but this increase was not statistically significant”, to avoid misinterpretation.
lines 149-150: Please leave these kind of speculations out and just state that there were no statistically significant differences.
lines 156-157: Please remove, unnecessary repetition.
lines 162-163: These sentence belongs to the discussion.
lines 165-169: Please rewrite: the content is observed not to change across light treaments except for BL. The last sentence belongs to discussion.
lines 180-181: As the effect was not statistically significant, omit the sentence.
lines 179-187: This paragraph continues with misinterpretation and/or reporting results only in part and/or not stating to which treatment one is comparing against. Please rewrite.

Throughout Discussion: Why are you referring to the specific studies you are citing? Please explain whether it is because the plant species were similar to the one you used or what was the reasoning. Please also vary the way how you write here, now you almost invariably start with e.g. Macedo et al. found that... This makes the text more like a list of studies, not discussion. Please go through the discussion after you have gone through the results and carefully check what you can say: if the effect was not statistically significant, then there were no decreases or increases.
lines 189-194: This paragraph should be in the introduction, not here.
line 195: Please remove the first sentence, it is redundant.
line 196: Please avoid using word “obviously”.
198-199: You simply can not phrase it like this, as there were no statistically significant differences between the light treatments and control.
line 209: “Believed” is not a correct verb to use in scientific article, either they demonstrated or showed something.
line 219: “Chlorophyll b is not sensitive to light quality.” Please rephrase to “Chlorophyll b was not sensitive to light quality in our study.” Or if you are referring to previous studies, references are needed.
lines 223-230: This paragraph should be in the introduction or materials & methods, where you justify your methodology.

Figure 1: It is necessary to have units in the y-axis, preferably µmol/m-2/s-1, and not W/m2 in this case. In addition, please make x-axis scale the same for each panel, to make comparison between light treatments easier.

Figure legends: You need to include the phrase “Values represent mean ± SE…” that you have now for Table 1 legend in each and every figure legend, as they need to be understandable on their own.

Please consider having Figures 6-9 and 11-12 as one figure with different panels of what you now have as separate figures, this would save space and it would make it easier to compare the measured responses and draw conclusions if some treatments affected certain similar responses in a similar manner.

---

## Round 0.2 · Minor Revisions

Dear author,

I have received the review reports from two external reviewers. You did a nice work editing your manuscript, but I think that there are several points requiring a further effort of edition before final acceptation. Please edit all points suggested by reviewer 1 and those from myself (see below). Please, when uploaded the reviewed manuscript, use blue font for the new text and last editions.

Comments from the academic editor:

My main concern is that your aim, and mostly your discussion, lacks of a putative mechanistic pathway linking your light treatments with the expected results. What are the physiological effects of monochromatic light on plants? Why monochromatic light supplementation is expected to produce this or other effects on plant physiology? Please, add some lines about this in the introduction and also in the discussion sections.

Besides, there is several minor points with the methodology that should be clarified:

L56. You refer here to light intensity, isn’t it?
L67. In instance, the height of …
L75. Be more precise than “rarely”. Were those effects previously studied or not?
L102. Be more precise with the replication. Do you mean three replicated trays, that is 3 x 110 plants for each treatment?
Methods: I would express the concentration of extraction solutions in a more common units, in instance vol:vol. Always water based dilutions?
L128. Please add the extraction ratio, I mean 0.5 g fresh weight in how much extraction solution?
L301:302. Delete this disconnected sentence.
Table 1. Reorganize this table putting the treatments up and the variables in the left column. The second column would be the F and P for the treatment effect on each variable. Add the df of the error term. Report as F(df effect, df error term) = ####; P = ####).
Fig 4, 5 and 6. Add the df of the error term
Fig 5. Edit “Fluorescence”

Reviewer 1 ·

Basic reporting

The revised manuscript entitled: ‘Effects of supplemental lighting with different light qualities on growth and secondary metabolite content of Anoectochilus roxburghii’ deserves publication in PeerJ - however, in my opinion, it still needs some revision. My specific comments/remarks are given below.

Experimental design

no comment

Validity of the findings

no comment

Additional comments

Generally
I recommend proofreading of the manuscript by a native English speaker.
Please make an effort to avoid editorial mistakes (e.g. lack of spacing, double dots, lack of consistency in formatting etc.).

Materials and methods
Is it really necessary to express the acetone concentration with such accuracy - 10.652 mol/l ? The same applies to sections entitled ‘Root activity’, ‘Total Flavonoides’ and ‘Total Polyphenols’.

Results
Please specify the units when you describe parameters, for example stem diameter, root length, chlorophyll content, etc.

Figure and Table Legends
Figure 4, 5, 6, 7, Table 1
In my opinion there is no need to write in the figure legends: ‘In the same row, values with the same superscript letter are not significantly different (P > 0.05); those with different superscript letters are significantly different (P < 0.05)’. This sentence should be written as follows: ‘In the same row, values marked with different letters are significantly different (P < 0.05)’. The same applies to the caption of Table 1.

References
Please, check all references once again because in this section are still mistakes (lack of spacing and italics, double dots, lack of consistency in formatting etc.).

·

Basic reporting

No comment.

Experimental design

No comment.

Validity of the findings

No comment.

Additional comments

The authors have clearly taken the time and effort and carefully made corrections and additions according to the comments and requests by the editor and the reviewers.

---

## Round 0.3 · accepted · Accept

Dear author,

I am sorry the delay in communicating my decision. I had some unexpected personal problems.Thanks for your last effort, I really think that the quality of your paper is greater after the revision process.
Thanks for submitting your research to PeerJ

Best wishes,

Luis Sampedro

#